# TRPM7 Modulates Human Pancreatic Stellate Cell Activation

**DOI:** 10.3390/cells11142255

**Published:** 2022-07-21

**Authors:** Julie Auwercx, Philippe Kischel, Thibaut Lefebvre, Nicolas Jonckheere, Alison Vanlaeys, Stéphanie Guénin, Silviya Radoslavova, Isabelle Van Seuningen, Halima Ouadid-Ahidouch, Hemant M. Kocher, Isabelle Dhennin-Duthille, Mathieu Gautier

**Affiliations:** 1UR-UPJV 4667, UFR des Sciences, Université de Picardie Jules Verne, F-80039 Amiens, France; julie.auwercx@etud.u-picardie.fr (J.A.); philippe.kischel@u-picardie.fr (P.K.); thibaut.lefebvre@u-picardie.fr (T.L.); alison.vanlaeys@etud.u-picardie.fr (A.V.); silviya.radoslavova@u-picardie.fr (S.R.); halima.ahidouch-ouadid@u-picardie.fr (H.O.-A.); isabelle.dhennin@u-picardie.fr (I.D.-D.); 2Université Lille, CNRS, Inserm, CHU Lille, UMR9020-U1277—CANTHER—Cancer Heterogeneity Plasticity and Resistance to Therapies, F-59000 Lille, France; nicolas.jonckheere@inserm.fr (N.J.); isabelle.vanseuningen@inserm.fr (I.V.S.); 3Centre de Ressources Régionales en Biologie Moléculaire, Université de Picardie Jules Verne, F-80039 Amiens, France; stephanie.vandecasteele@u-picardie.fr; 4Centre for Tumour Biology, Barts Cancer Institute—A CRUK Centre of Excellence, Queen Mary University London, London EC1M 6BQ, UK; h.kocher@qmul.ac.uk

**Keywords:** pancreatic stellate cells, cancer, TRPM7

## Abstract

Pancreatic diseases, such as pancreatitis or pancreatic ductal adenocarcinoma, are characterized by the presence of activated pancreatic stellate cells (PSCs). These cells represent key actors in the tumor stroma, as they actively participate in disease development and progression: reprograming these PSCs into a quiescent phenotype has even been proposed as a promising strategy for restoring the hallmarks of a healthy pancreas. Since TRPM7 channels have been shown to regulate hepatic stellate cells proliferation and survival, we aimed to study the role of these magnesium channels in PSC activation and proliferation. PS-1 cells (isolated from a healthy pancreas) were used as a model of healthy PSCs: quiescence or activation were induced using all-trans retinoic acid or conditioned media of pancreatic cancer cells, respectively. The role of TRPM7 was studied by RNA silencing or by pharmacological inhibition. TRPM7 expression was found to be correlated with the activation status of PS-1 cells. TRPM7 expression was able to regulate proliferation through modulation of cell cycle regulators and most importantly p53, via the PI3K/Akt pathway, in a magnesium-dependent manner. Finally, the analysis of TCGA database showed the overexpression of TRPM7 in cancer-associated fibroblasts. Taken together, we provide strong evidences that TRPM7 can be considered as a marker of activated PSCs.

## 1. Introduction

Pancreatic stellate cells (PSCs) represent 4 to 7% of the healthy pancreas, and are normally in a quiescent state. Their physiological roles are mainly to store retinoids into lipid droplets, and to regulate the extracellular matrix (ECM) homeostasis by secreting metalloproteinases as well as their inhibitors [1]. Under pathological conditions, such as pancreatitis or cancer, PSCs change to an activated myofibroblast-like phenotype, leading to enhanced cell migration and proliferation [2]. Importantly, an intense ECM deposition from the activated PSCs is a distinctive feature of pancreatic ductal adenocarcinoma (PDAC), one of the deadliest cancers [3]. This stromal remodelling—primarily driven by the activation of PSCs—is called desmoplasia, and is responsible for metastasis formation and chemoresistance commonly observed in PDAC [4]. PSCs are not only involved in fibrosis and stroma remodelling as they can also interact with exocrine cells. Thus, PSCs may regulate exocrine function through the cholecystokinin-induced release of acetylcholine, but in a modest manner [5]. These data are still under debate as some studies showed that PSCs blocks amylase secretion by acinar pancreatic cells [6], as well as Ca^2+^-associated signalling [7] in co-culture models. Finally, it has been shown that PSC activation allows exocrine but not endocrine cell differentiation in the developing pancreas [8]. Recently, pentraxin 3, which is specifically secreted by activated PSCs, has been identified as a stromally derived biomarker for PDAC [9]. Moreover, it is now well established that activated PSCs can communicate with cancer cells through cytokines and growth factor secretion [10]. Targeting activated PSCs has been proposed as a promising strategy against PDAC progression. Although stroma depletion induces undifferentiated and more aggressive cancer, a strategy based on the reprogramming of activated PSCs toward a quiescent phenotype could restore the hallmarks of a healthy pancreas without degrading key components needed for tissue homeostasis [11]. For example, calcitriol, the ligand of the vitamin D receptor, potentiates the antitumoral effect of gemcitabine by inducing quiescence of PSCs [11]. More recently, treatment with all-*trans*-retinoic acid (ATRA) has been shown to limit the desmoplasia and to inhibit PSC proliferation by restoring their quiescent phenotype, providing a promising complement to gemcitabine-nab-paclitaxel chemotherapy [12]. However, PDAC is also characterized by a stromal heterogeneity due to the presence of multiple cell subtypes, which may explain the differential response to chemotherapies [13]. Therefore, a better understanding of the mechanisms controlling phenotype transition of PSCs will help to identify new biomarkers of PSC activation and to develop new therapeutic tools against PDAC.

Ion channels are integral proteins that are involved in almost all cellular processes. We and others have shown that the Transient Receptor Potential Cation Channel Subfamily M Member 7 (TRPM7) is overexpressed in PDAC and its expression is correlated with a poor survival [14,15,16]. TRPM7 is a non-selective cation channel fused with a functional kinase located in the C-terminus [17,18]. TRPM7 is a dual-function protein. The pore domain allows the entry of Zn^2+^, Ca^2+^ and Mg^2+^ cations as well as some toxic metals such as Cd^2+^ or Ni^2+^ [19]. TRPM7 is involved in the intestinal absorption of divalent cations [20] and the regulation of Mg^2+^ cell homeostasis [21]. On the other hand, the kinase domain belongs the α-kinase family and its activity is distinct from the activity of the channel [22]. While the physiological role of TRPM7 kinase domain is far to be understood, some in vitro substrates have been identified to regulate various cellular processes such as adhesion, migration or gene expression (for review see [23]). This protein is essential for cell survival and development [24]. Moreover, TRPM7 is the main gatekeeper of free magnesium (Mg^2+^) intake [20,21]. One of the particularities of the Mg^2+^ cation is its almost equal distribution across the plasma membrane (around 1 mM). Consequently, there is almost no electrochemical gradient for Mg^2+^ transport contrarily to the others cations (Ca^2+^, Na^+^, K^+^). However, there are strong evidences showing that Mg^2+^ channels such as TRPM6 and TRPM7 allow Mg^2+^ influx into the cell [25]. High intracellular levels of Mg^2+^ have been observed in transformed cells giving them a metabolic advantage to proliferate [26]. Interestingly, it has been shown that TRPM7 expression regulates hepatic stellate cell (HPC) proliferation and survival [27,28].

The aim of this study was to assess the role of TRPM7 in normal PSC activation. Indeed, we hypothesized that TRPM7 could be involved in the activation of PSCs by regulating intracellular Mg^2+^ and cell proliferation. We therefore studied the expression of TRPM7 in both models of quiescent and activated PSCs. The role of TRPM7 on PSC proliferation was then assessed by silencing and pharmacological approaches. We provide strong evidences that the TRPM7 channel was able to regulate PSC proliferation by affecting p53 via the PI3K/Akt pathway in a magnesium-dependent manner.

## 2. Materials and Methods

### 2.1. Cell Lines and Treatments

Experiments were conducted on PS-1, an immortalized human PSC line, isolated from a healthy pancreas [29]. RLT-PSCs, an immortalized human PSC line isolated from chronic pancreatitis, were used as a model of pathological activated PSCs [30]. Both PS-1 and RLT-PSCs were grown in DMEM/F12 media (Gibco^®^, Villebon-sur-Yvette, France) supplemented with 10% fetal bovine serum (FBS) GOLD (PAA Laboratories) and selected with 1µg/mL puromycin (Gibco^®^, Villebon-sur-Yvette, France). Cancer-cell-conditioned media were prepared by using MIA PaCa-2 pancreatic cancer cell line (CRL-1420^TM^ ATCC^®^, Manassas, VA, USA). Cells were cultured at 37 °C in a humidified atmosphere with 5% CO_2_, the media were changed every 48 h and the cells were trypsinized once a week to prepare the experiments. Cells were treated with pharmacological inhibitors as indicated. All-Trans-Retinoic Acid, Nile Red, NS8593, nutlin-3A and wortmannin were purchased from Sigma Aldrich (Saint-Quentin-Fallavier, France).

### 2.2. Cell Transfection

Cells were transfected with siRNA using a Nucleofector^TM^ II device (Lonza, Colmar, France) as previously described [31]. Cells were transfected with a siRNA targeting TRPM7 mRNA (siTRPM7: 5′-GUCUUGCCAUGAAAUACUCUU-3′), or with a scrambled siRNA used as a control (siControl: 5′-CUGGACAUGGACCAAGUGGACUU-3′). Both siRNAs were purchased from Eurogentec (Seraing, Belgium).

### 2.3. RT-qPCR

RNA extraction and quantitative RT-PCR experiments were performed as previously described [32]. The primer sequence for human TRPM7 were sense 5′-GTCACTTGGAAACTGGAACC-3′ and anti-sense 5′-CGGTAGATGGCCTTCTACTG-3′, giving an amplified RT-PCR product of 273 bp. The other primer sequences used were: TRPM6 sense 5′-GAGGAGATGGATGGGGGC-3′ and anti-sense 5′-GGTCCAGTGAGAGAAAGCCAA-3′ (413 bp), GAPDH sense 5′-AGGGGCCATCCACAGTCTTC-3′ and anti-sense 5′-AGAAGGCTGGGGCTCATTTG-3′ (165 bp). The ratios of TRPM6 and TRPM7 mRNAs were normalized by GAPDH mRNA in each sample.

### 2.4. Immunoblotting

Proteins were extracted from cell lysates and detected by Western blots as previously described [31]. Briefly, equal amounts of each protein sample (50 µg) were separated by electrophoresis on sodium dodecyl sulfate (SDS) polyacrylamide gel electrophoresis and blotted onto nitrocellulose membrane (Amersham, Cardiff, UK). Blots were incubated with primary antibodies (Table 1) and developed with the enhanced chemiluminescence system using specific peroxidase-conjugated anti-IgG secondary antibodies.

### 2.5. Cell Viability Assays

We used the MTT ((3-(4,5-dimethylthiazol-2-yl)-2,5-diphenyl tetrazolium bromide) assay which is a colorimetric assay for cell survival and proliferation [33]. Only living cells with active mitochondria are able to degrade MTT into formazan dark blue crystals. The amount of formazan generated per cell depends on the level of energy metabolism in the cell. As only living cells are metabolically active, MTT is an indirect measurement of cell viability. In order to better assess the cell proliferation, we completed the MTT assay with an analysis of the cell cycle by flow cytometry.

### 2.6. Cell Cycle Analysis

Cell cycle analysis was conducted using DNA cellular content quantification by flow cytometry. 1 × 10^6^ PS-1 cells were fixed with ethanol and sorted out by a cytometer (Accuri C6, BD Biosciences, Rungis, France). The percentage of cells in each phase of the cell cycle was obtained by calculating the area under the curve using the Cyflogic version 1.2.1 software (CyFlo Ltd., Turku, Finland).

### 2.7. Electrophysiological Recordings

TRPM7 channel activity was assessed by recording the Magnesium Inhibited Cation (MIC) current using the conventional patch-clamp technique in the whole-cell configuration as previously described [32]. Briefly, membrane potential was held to −40 mV and currents were elicited by a ramp depolarization from −100 to +100 mV for 350 milliseconds at a frequency of 0.1 Hz. The MIC currents developed after the dialysis of intracellular media by the free Mg intrapipette solution [34]. The current–voltage (I–V) relationship was built as the difference between the steady-state current activated by the depletion of [Mg^2+^]_i_ and the basal current recorded few minutes after the patch rupture. Currents were expressed as current densities (in pA.pF^−1^) by dividing the current intensity (in pA) by the cell capacitance (in pF). All experiments were performed at room temperature.

### 2.8. Cell Imaging Experiments

PS-1 cells were plated on glass coverslips in 35 mm diameter dishes at a density of 2.5 × 10^5^ cells. After 48 h, cells were loaded in cell growth medium at +37 °C for 45 min with 3 µM Fura-2-AM (Sigma-Aldrich, Saint-Quentin-Fallavier, France) or with 3 µM Mag-Fura-2-AM (Invitrogen, Thermo Scientific™, Illkirch, France). We used the technique of Mn^2+^ quenching protocol to evaluate the constitutive divalent cation entry, as previously described [31,32]. The basal cytosolic Ca^2+^ level was estimated as the ratio of Fura-2 fluorescence intensities measured with excitation at 350 and 380 nm (F350/F380). The basal cytosolic Mg^2+^ level was estimated as the ratio of Mag-fura-2 fluorescence intensities measured with excitation at 330 and 370 nm (F330/F370).

### 2.9. Analysis of CAF Datasets

Analyses of relative expression of *TRPM7*, *TRPM6* and *POSTN* were conducted using GEPIA tool (http://gepia.cancer-pku.cn/, accessed the 2 March 2022) that integrates a normal pancreas from GTEx (*n* = 171) and pancreatic cancer PAAD dataset from TCGA (*n* = 179). Expression of *TRPM7*, *TRPM6* and *POSTN* in different cell types were analyzed using GEPIA2021 and EPIC deconvolution tools (http://gepia2021.cancer-pku.cn/, accessed the 2 March 2022) [35,36]. Quantitative comparisons of the cell proportions or expression in different cell types were analyzed using built-in t test (GEPIA) or ANOVA test (GEPIA2021). The GSE28735 dataset contains 45 normal pancreases (adjacent non tumoral, ANT) and 45 tumor (T) tissues from PDAC cases. mRNA levels were analyzed using the National Center for Biotechnology Information (NCBI) Gene Expression Omnibus (GEO) database (http://www.ncbi.nml.nih.gov/geo/, accessed the 2 March 2022).

### 2.10. Statistical Analysis

Data are presented as Mean ± S.E.M. and the number of separate experiments or the number of studied cells is represented by *n*. Experiments were repeated in at least three different cell passages. Data analysis and figure conception were made by using Microcal™ Origin^®^ (OriginLab Corporation, Northampton, MA, USA), Clampfit (Molecular Devices, Inc., San José, CA, USA) and Inkscape software. Statistical analyses were made using Student’s –test or Mann–Whitney rank sum test depending on sample normality determined by paired Wilcoxon signed rank test. A two-way analysis of variance was used followed by the Holm–Sidak method to compare two parameters or more. All the statistical analyses were conducted using Sigma-Stat 3.0 (Inpixon, Palo Alto, CA, USA).

## 3. Results

### 3.1. TRPM7 Is a Marker of Human Pancreatic Stellate Cell Activation

All-*trans* retinoic acid (ATRA) has previously been shown to induce quiescence in PSC cells, notably increasing fat-containing droplets and reducing cell viability [37]. In PS-1 cells (isolated from a normal pancreas [29]), more lipid droplets were indeed detected using Nile red dye (1:1000 for a minimal of 30 min) following ATRA treatment (Figure 1A). At the molecular level, ATRA was found to decrease the expression of vimentin (Figure 1B), as previously reported [38]. PS-1 cell viability was also decreased after ATRA treatment (Figure 1C). In these more quiescent stellate cells, TRPM7 protein expression was found to be reduced (Figure 1D), and this reduction was confirmed at the mRNA level (Figure 1E). We then tried to activate PS-1 cells by exposing them to MIA-PACA-2 cancer-cell-conditioned medium. With this latter treatment, cell viability was significantly increased (Figure 1F). Most importantly, TRPM7 protein expression level was increased by a factor of 2 (Figure 1G), and this increase was also seen at the mRNA level (Figure 1H). We then wondered whether TRPM7 could be actually a marker of human pancreatic stellate cell activation by measuring expression levels in another pancreatic stellate cell line, named RLT-PSC, isolated from patients with chronic pancreatitis [30]. In these activated PSC, cell viability was found largely increased when compared with PS-1 cells (Figure 1I). In this chronic pancreatitis model, expression of TRPM7 was also increased at the protein level (Figure 1J).

### 3.2. TRPM7 Regulates Cell Proliferation through Modulation of Cell Cycle Regulators

#### 3.2.1. TRPM7, Cell Proliferation and Cell Cycle Analysis

To assess the role of TRPM7 in PS-1 cells, we used a siRNA strategy to selectively inhibit TRPM7 expression, as previously carried out in human pancreatic cancer cells [15,33,39]. TRPM7 silencing decreased TRPM7 mRNA (Appendix A) without affecting *TRPM6* mRNA expression (Appendix A). The inhibition of TRPM7 protein expression by siRNA was confirmed by immunoblotting (Figure 2A). Interestingly, TRPM7 silencing significantly reduced cell viability after 72 h and 96 h of cell growth (Figure 2B). This effect was accompanied with an accumulation of cells in the G0/G1 phase of the cell cycle, with a concomitant decrease in the number of cells in the S and G2/M phases (Figure 2C). The role of TRPM7 on PS-1 cell proliferation was also assessed by blocking the channels with NS8593, a pharmacological blocker of TRPM7 channels [40]. Overall, 48 h treatment with NS8593 (25µM) almost fully abolished cell growth, with a decrease in cell viability at 72 and 96 h (Figure 2D). As for TRPM7 silencing, NS8593 increased the percentage of cells in G0/G1 and decreased the percentage of cells in S and G2/M phases of the cell cycle (Figure 2E). Taken together, these results showed that TRPM7 inhibition, using either a siRNA or a pharmacological approach, strongly altered the cell proliferation by causing an accumulation of PS-1 PSCs in the G0/G1 phase of the cell cycle.

#### 3.2.2. TRPM7 Silencing and Expression of Cell Cycle Regulators

Since TRPM7 silencing or inhibition reduced viability, we then studied the effect of TRPM7 silencing on the expression of proteins involved in G1/S transition of the cell cycle. Cyclin-E, p53, CDK2, and PCNA expressions were assessed by Western blot (Figure 3A,B). p53 expression was increased (Figure 3A,B), whereas Cyclin-E, CDK2 and PCNA expressions were decreased following TRPM7 silencing (Figure 3A,B). Collectively, these data showed that TRPM7 inhibition decreased PS-1 cell proliferation by cell cycle arrest in G0/G1 phase notably by affecting p53, Cyclin-E, CDK2 and PCNA expression levels.

To assess the p53 involvement, a 24 h treatment with Nutlin-3a (20 µM, a potent inhibitor of Mdm2-p53 interactions leading to p53 stabilization [41]) was tested on PS-1 cell proliferation (Figure 3C–F). Nutlin-3a decreased PS-1 viability at 72 h and 96 h (Figure 3C) and the number of cells in the S phase of the cell cycle (Figure 3D). The activation of p53 pathway by Nutlin-3a was confirmed by the increase in p53 expression (Figure 3E,F). Moreover, Nutlin-3a treatment induced a decrease in both CDK2 and PCNA expressions (Figure 3E,F), suggesting that TRPM7 would be able to affect p53 stability, which in turn affects cell cycle regulators. Finally, Nutlin-3a decreased TRPM7 expression, suggesting that p53 was able to regulate TRPM7 expression in PSCs (Figure 3E,F), most probably by direct interaction with the numerous p53 responsive elements found in the TRPM7 gene (8 according to p53famTag [39]).

### 3.3. How Does TRPM7 Regulate Proliferation?

We next wondered how TRPM7 was able to influence proliferation. It has been shown that TRPM7 regulates hepatic stellate cell proliferation through the PI3K and ERK pathways [42]. To study the involvement of the PI3K pathway in PS-1 cell proliferation, 24 h treatment with wortmannin (1 µM) was first used to selectively inhibit PI3K. Wortmannin was actually found to inhibit PS-1 cell viability at 72 and 96 h (Figure 4A). Wortmannin treatment induced the accumulation of cells in G0/G1 phase of the cell cycle and decreased the proportion of cells in S and G2/M phases (Figure 4B). Next, we sought to determine whether TRPM7 inhibition had any effect on PI3K pathway. Interestingly, the pAkt/Akt and the pMdm2/Mdm2 ratios were decreased by TRPM7 silencing (Figure 4C,D). On the other hand, TRPM7 silencing was not able to modify the pERK/ERK ratio in PS-1 cells (Figure 4C,D). Collectively, our data demonstrated that PI3K was implicated in PS-1 cell proliferation, and that TRPM7 inhibition was able to affect the PI3K/Akt pathway but not ERK pathway.

### 3.4. TRPM7 Regulates Proliferation through the Akt Pathway: Is the Process Calcium or Magnesium-Dependent?

TRPM7 is a non-selective cation channel which is required for cellular Mg^2+^ homeostasis [25]. The functional expression of TRPM7 in PS-1 pancreatic stellate cell plasma membrane was assessed by whole-cell patch-clamp and by Mn^2+^ quenching recordings (Figure 5A). Mg^2+^ Inhibited Cation (MIC) currents are induced by dialyzing the intracellular media by a Cs^+^ intrapipette solution containing EGTA in order to chelate intracellular Mg^2+^, as previously described [43]. Dialysis by EGTA induced the development of large outwardly rectifying currents in PS-1 cells (Figure 5A). I-V curves displayed the typical characteristics of MIC currents, with reversal potential close to 0 mV and outward rectification for positive membrane potentials (Figure 5B). The dominating role of TRPM7 channel in MIC current generation was confirmed by the inhibition of these currents by TRPM7 silencing (Figure 5A,B). Next, Mn^2+^ quenching recordings were used to study the role of TRPM7 in Mn^2+^ basal entry. TRPM7 silencing reduced Mn^2+^ quench slopes when compared to scrambled siRNA (Figure 5C,D). Treatment with NS8593 induced a larger decrease in Mn^2+^ quench slopes (Figure 5E,F). Taken together, these results strongly suggest the implication of TRPM7 in the basal entry of divalent cations in PSCs. Since both calcium and magnesium can enter the cells via TRPM7, we sought to determine which cation was involved in the physiological effects seen on pancreatic stellate cells: (1) when TRPM7 is silenced, (2) when TRPM7 is pharmacologically inhibited, and (3) when PSCs return to quiescence. TRPM7 silencing, NS8593 or ATRA treatments significantly and similarly decreased the MagFura-2 basal fluorescence ratio in PSCs (Figure 5G). On the other hand, the basal fluorescence ratio of the intracellular Fura-2 calcium probe was slightly increased following TRPM7 silencing or ATRA treatment, whereas no effects were observed on Fura-2 fluorescence after NS8593 treatment (Figure 5H).

These results suggest that magnesium was, by far, the best candidate to explain the physiological effects observed in PSCs. In good agreement with this, deprivation of Mg^2+^ in the culture media induced a decrease in cell viability (Figure 6A) associated with an accumulation in the G0/G1 phase and a decrease in cell numbers in S and G2/M phases (Figure 6B). Mg^2+^ deprivation induced similar effects than TRPM7 inhibition or ATRA treatment. Conversely, Mg^2+^ supplementation reversed the effect of TRPM7 silencing on cell cycle (Figure 6D), although the effect on cell viability was modest (*p* < 0.07; Figure 6C). Moreover, Mg^2+^ supplementation reversed the effect of TRPM7 silencing on p53 expression (Figure 6E,F).

### 3.5. TRPM7 Expression Is Increased in Cancer-Associated Fibroblasts (CAFs)

We investigated the expression of TRPM7, TRPM6 and periostin (POSTN), a marker of activated CSPs [13,44], in pancreatic cancer samples using GEPIA tool. We observed that *POSTN* mRNA relative level was increased in tumor samples when compared with a normal pancreas (*p* < 0.05, Figure 7A). We also observed a trend toward an increase in *TRPM7* that was not statistically significant. *TRPM6* expression was not altered in PDAC. Spearman correlation analysis showed that *TRPM7* and *POSTN* were positively correlated in tumor samples (r = 0.24; *p* = 0.0013, Figure 7B). We also observed similar findings in an independent PDAC dataset (GSE28735), in which *POSTN* mRNA level was increased in tumor sample compared to adjacent non tumoral tissue (not shown). Moreover, we showed a correlation between *POSTN* and *TRPM7* (r = 0.4041; *p*= 0.0059, Figure 7C). As these samples were originated from surgical resection, the intrinsic intra-tumoral heterogeneity also led to high level of stromal cells, including cancer-associated fibroblasts. We investigated the expression of *TRPM7*, *TRPM6* and *POSTN* in pancreatic cancer samples (PAAD-TCGA) and normal pancreas samples from GTEX using GEPIA2021, which enables to filter the cell types of interest in normal tissue from GTEX or cancer types from TCGA [36]. The expression was analyzed in CD4+, CD8+, endothelial cells, macrophages and cancer associated fibroblasts (CAFs) using EPIC deconvolution tool [35]. PAAD samples are enriched with CAFs, endothelial and immune cells (not shown). We observed that *TRPM7* and *POSTN* expression is significantly enriched in CAFs (*p* < 10^−15^) and to a lower extent in endothelial cells from PAAD tumor samples compared with a normal pancreas from GTEX (Figure 7D). *TRPM6* was not detected in any cell type analyzed.

## 4. Discussion

Pancreatic ductal adenocarcinoma (PDAC) is characterized by an abundant desmoplastic stroma. This tumor stroma is constituted by ECM proteins as well as various cell types, including cancer-associated fibroblasts (CAF), endothelial and immune cells [45]. CAFs are able to re-shape the microenvironment through ECM remodeling and interactions with the cellular compartment [46]. In PDAC, CAFs mostly originate from pancreatic stellate cells (PSCs) [13]. In their quiescent state, these PSCs are mostly characterized by lipid droplets containing vitamin A [4,5], and also characterized by vimentin expression. Upon activation, for instance, when engaged in inflammatory states or cancer, PSCs are turned into an activated phenotype, characterized among others by the loss of lipid droplets. The study presented herein is based on two cell lines: PS-1 cells and RLT-PSC cells. While PS-1 are pancreatic stellate cells isolated from a normal human pancreas, they are certainly not completely quiescent cells, as PSCs undergo activated phenotypic transition in only 48 h when cultured [29]. PS-1 are thus neither completely quiescent, nor completely activated cells. Retinol and one of its metabolites, ATRA, induce quiescence in cultured PSCs [37]. Upon exposure to ATRA, PSCs quiescence is usually associated with increased lipid droplets, but also decreased proliferation rates [37,47], as well as a decrease in vimentin expression [29]. Our results are in good agreement with this: PS-1 returned to quiescence upon ATRA treatment, whereas untreated PS-1 cells were activated through incubation with MIA-PaCa-2 (pancreatic cancer cells) conditioned medium. We show that the TRPM7 channel expression was correlated with the PS-1 activation state. TRPM7 expression was investigated because this channel was shown to be implicated in apoptosis of hepatic stellate cells [48]. Since TRPM7 expression was also in line with the activation status of another cell line, namely, RLT-PSC cells (established from tissue obtained from a chronic pancreatitis tissue resection), our results strongly suggest that TRPM7 is a marker of human pancreatic stellate cell activation.

Since proliferation was affected by activation status, and that the proliferation status was correlated with TRPM7 expression, we tried to check whether TRPM7 expression was only modulated as a consequence of activation, or whether TRPM7 was itself one of the causes of activation. We have shown that vimentin was affected by TRPM7 silencing (about 50% increase, Appendix A). TRPM7 expression and/or function was able to recapitulate the activation-dependent effects seen on proliferation in PS-1 but also in RLT-PSCs (Appendix A). Our results indeed showed a reduction in viability, not only with selective silencing, but also with pharmacological inhibition of TRPM7. This silencing was shown to be caused notably by an accumulation of the cells in the G0/G1 phase of the cell cycle (with a simultaneous decrease in the percentage in S phase). These results are in good agreement with the literature: TRPM7 expression increases dramatically during the G1 phase, suggesting its implication in the transition to the S phase [49]. It has also been shown that TRPM7 was able to regulate G1-S transition in retinoblastoma cells [50] and bladder cancer cells [51]. We additionally showed that TRPM7 silencing (and the subsequent inhibition of the G1-S transition) was accompanied by an increase in p53 expression. This latter was able to reduce cyclin E, CDK2 and PCNA expression. It is noteworthy that if TRPM7 expression was able to modulate p53 expression, p53 expression was able to modulate TRPM7 expression, as shown with the use of nutlin-3a (Figure 3). This is likely by direct interaction with the eight p53 responsive elements that are present in the TRPM7 gene (found with p53famTag [39]).

We additionally discovered that TRPM7 was able to regulate PS-1 proliferation through the PI3K/Akt pathway. Wortmannin, by inhibiting PI3K, was able to recapitulate the effects seen on cell viability and on the cell cycle (Figure 4C). The effect of TRPM7 silencing was a decreased pAkt/Akt ratio, leading to a reduced pMdm2/Mdm2 ratio. This is in good agreement with the work of Fang and collaborators, in which TRPM7 was shown to regulate proliferation of hepatic stellate cells through PI3K and cell cycle proteins (PCNA, Cyclin D1 and CDK4) [42]. On the other hand, we did not see any effect of TRPM7 inhibition on the pERK/ERK ratio. This constitutes a difference with what was described in hepatic stellate cells, where TRPM7 channel was shown to regulate PDGF-BB-induced proliferation via both PI3K and ERK pathways [42].

TRPM7 is known to regulate cation influxes through the plasma membrane: both calcium and magnesium are concerned, and these cations are of note able to regulate cell proliferation [52]. It was thus mandatory to check whether the PSC proliferation was regulated by calcium or magnesium in PS-1 cells. Our results from patch-clamp and cation imaging confirmed that TRPM7 was indeed an actor of cation influxes, as it was able to specifically regulate intracellular magnesium concentration in PS-1 cells. Reduction in magnesium entry through TRPM7 silencing also confirmed some membrane location of the channel in these PS-1 cells. It has also been shown that when TRPM7 expression decreases, the expression of other magnesium transporters increases to maintain magnesium homeostasis in the cell: this is notably the case for MagT1 [53]. Such a mechanism is apparently not present in PS-1 cells, or at least, the compensation is at best partial. These results showed that TRPM7 channel was the main actor of magnesium homeostasis in PS-1 cells. However, MagFura-2 probe has been also used to monitor Ca^2+^ release from ER stores [54]. Moreover, TRPM7 currents are required for refilling ER Ca^2+^ contents at rest and after depletion DT40 B lymphocytes [55] and we cannot fully exclude the possibility of Ca^2+^ ER regulation by TRPM7 in PSCs. The assessment of such mechanism will need further investigations. Furthermore, we also showed that PSC proliferation was magnesium-dependent. Although magnesium deprivation decreased PSC viability, magnesium supplementation does not fully restore the effect of TRPM7 silencing. This was mainly due to a proapoptotic effect of magnesium (data not shown). On the other hand, our data strongly suggest that p53 expression is regulated by magnesium but the role of this cation in the PI3K/Akt pathway needs further investigations. Moreover, our results are obtained in immortalized cell lines and need to be confirmed in animal models and human tissues as it is known that there could be some discrepancies between cell line behaviour and in vivo models as previously shown for PSCs [56]. From a more general point of view, pancreatic diseases are initiated by the interplay between PSCs, immune cells and acinar cells as recently reviewed by Petersen et al. [57]. In particular, exposure to alcohol and fatty acids induced necrosis and release of soluble factors from acinar cells which in turn activated PSCs through bradykinin receptor activation. Interestingly, it has been shown that bradykinin increased TRPM7 expression in vascular smooth muscle cells [58] and in hepatocellular carcinoma cells [59] leading to enhanced cell migration. Thus, it is tempting to speculate that TRPM7 could be involved in the initial PSC activation leading to pancreatitis, which is a one of the major risks for pancreatic carcinogenesis. Figure 8 recapitulates the findings of the present study.

To summarize, our data indicated that: (i) the TRPM7 channel was able to regulate cation entry and magnesium homeostasis in activated human PSCs; (ii) the expression level of the TRPM7 channel varies according to the level of activation of the PSCs, a strong expression of TRPM7 being correlated with a high activation level; (iii) the TRPM7 channel was involved in the proliferation of activated human PSCs, by allowing the G1-S transition (mediated notably by CDK2 and PCNA) via the activation of the PI3K/AKT pathway; (iiii) this latter pathway was able to inhibit the p53 transcription factor in a magnesium-dependent manner. Taken together, our results strongly suggest that TRPM7 was involved in PSC activation, leading to enhanced proliferation.

Since pancreatic stellate cells (PSCs) are critical components and actors in the desmoplastic response, in the pro-survival and the pro-invasive features through multiple signaling cascades, targeting the tumor-promoting cancer–stromal cell interactions (i.e., normalizing the desmoplastic stroma) may enhance the effectiveness of conventional therapies: in this context, ATRA has recently been proposed as a stromal targeting agent for pancreatic cancer in a phase I clinical trial [12]. Since TRPM7 expression has a direct impact on the activation process of the PSCs, TRPM7 channel is of utmost importance in these cells and future studies will help define the role of TRPM7 modulation in in vivo experiments and as a potential target in pancreatic cancer.

## Figures and Tables

**Figure 1 cells-11-02255-f001:**
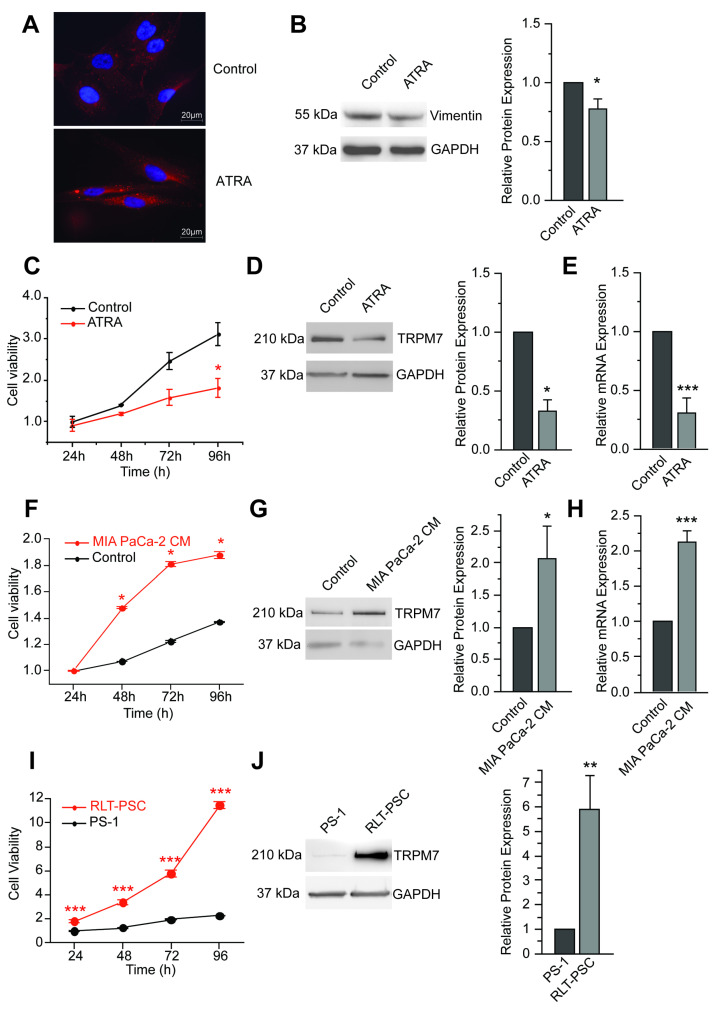
Correlation between TRPM7 expression and PS-1 activation status. (**A**) Lipid droplets were increased after ATRA treatment (as assessed by Nile Red staining). (**B**) Western blot analyses and associated histograms showed a reduced vimentin expression (*n* = 3). * indicates *p* < 0.05 (Mann–Whitney rank sum test). (**C**) Viability of PS-1 stellate cells was decreased upon ATRA treatment (*n* = 4). * indicates *p* < 0.05 (2-ways ANOVA). (**D**) TRPM7 protein expression was decreased in ATRA treated-cells (*n* = 3). * indicates *p* < 0.05 (Mann–Whitney rank sum test). (**E**) mRNA expression levels analyses showed a reduction in *TRPM7* transcription following ATRA treatment (*n* = 7). *** indicates *p* < 0.001 (Mann–Whitney rank sum test). (**F**) Viability of PS-1 stellate cells was increased when cells were activated by incubation with Mia PaCa-2 conditioned medium (*n* = 3). * indicates *p* < 0.05 (2-ways ANOVA). (**G**) TRPM7 protein expression was increased when PS-1 cells were cultured in Mia PaCa-2 conditioned medium (*n* = 4). * indicates *p* < 0.05 (Mann–Whitney rank sum test). (**H**) mRNA expression levels analyses revealed an increase in *TRPM7* transcription after incubation with Mia PaCa-2 conditioned medium (*n* = 3). *** indicates *p* < 0.001 (Mann–Whitney rank sum test). (**I**) Comparison between PS-1 and RLT-PSC stellate cells showed that RLT-PSC had higher viability than PS-1 cells (*n* = 4). *** indicates *p* < 0.001 (2-ways ANOVA). (**J**) Western blot showing TRPM7 expression in RLT-PSC cells and associated quantification (*n* = 6). ** indicates *p* < 0.01 (Mann–Whitney rank sum test).

**Figure 2 cells-11-02255-f002:**
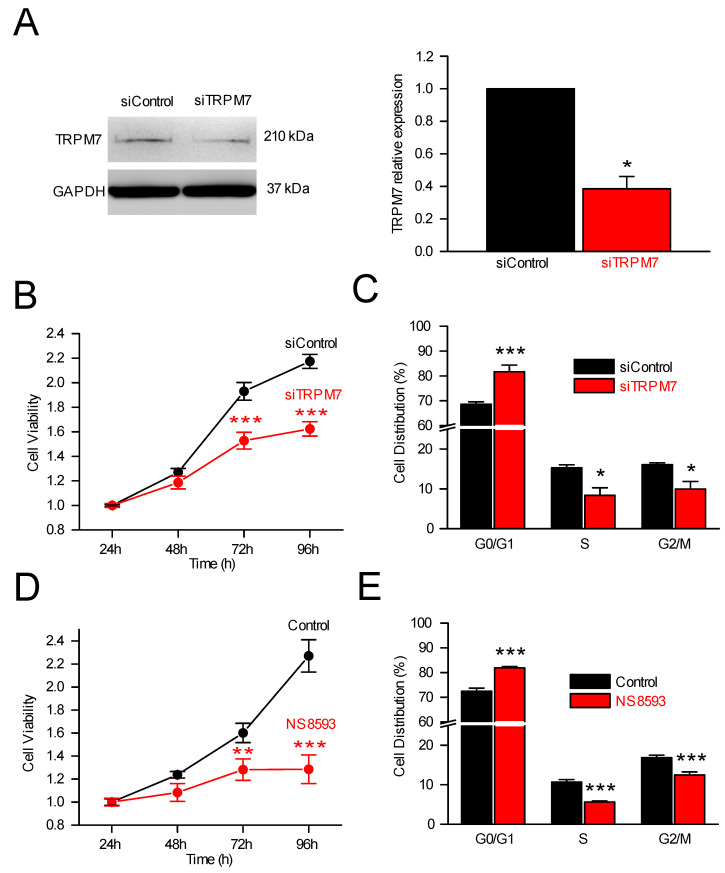
Proliferation was influenced by TRPM7 expression. (**A**) Immunoblotting of lysates of PS-1 cells transfected with a scramble siRNA (siControl) or a siRNA targeting *TRPM7* (siTRPM7), and incubated with anti-TRPM7 antibody (*n* = 4). * indicates *p* < 0.05 (Mann–Whitney Rank Sum Test). (**B**) PSC viability was decreased at 72 h and 96 h following TRPM7 silencing (*n* = 4). *** indicates *p* < 0.001 (2-ways ANOVA). (**C**) TRPM7 silencing led to enrichment of cells in G0/G1 while decreasing the percentages of cells in S and in G2/M phases of the cell cycle (*n* = 3). * indicates *p* < 0.05 and *** indicates *p* < 0.001 (2-ways ANOVA). (**D**) The pharmacological blocker of TRPM7 channels, NS8593 (25 µM), decreased PS-1 cell viability at 72 and 96 h (*n* = 3). ** indicates *p* < 0.01 and *** indicates *p* < 0.001 (2-ways ANOVA). (**E**) Following NS8593 treatment, PS-1 cells accumulated in G0/G1 phase, whereas the percentage of cells decreased in S and in G2/M phases (*n* = 3). *** indicates *p* < 0.001 (2-ways ANOVA).

**Figure 3 cells-11-02255-f003:**
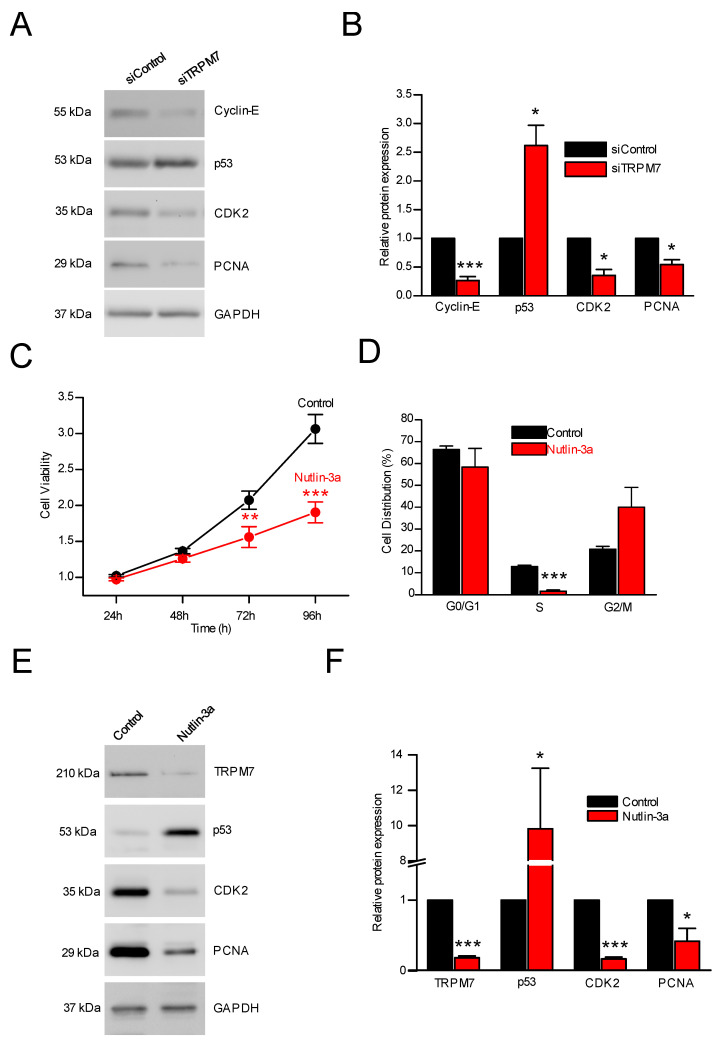
Influence of TRPM7 on cell cycle regulators (cyclin E, p53, CDK2, and PCNA expressions). (**A**) Immunoblotting of PS-1 protein lysates. The cells were transfected with a scramble siRNA (siControl) or a siRNA targeting TRPM7 (siTRPM7) and incubated with the following antibodies: anti-Cyclin-E, anti-p53, anti-CDK2, anti-PCNA, and anti-GADPH. (**B**) Quantification of Cyclin-E, p53, CDK2, and PCNA protein expression in PS-1 cells transfected with a scramble siRNA (siControl) or a TRPM7 targeting siRNA (siTRPM7) (*n* = 4–8). * indicates *p* < 0.05 and *** indicates *p* < 0.001 (Mann-Withney Rank Sum Test). (**C**) Cell viability without (Control) or with Nutlin-3, an inhibitor of the main p53 regulator, Mdm-2 (*n* = 4). ** indicates *p* < 0.01 and *** indicates *p* < 0.001 (2-ways ANOVA). (**D**) Following nutlin-3 treatment, the percentage of PS-1 cells was decreased in S phase (*n* = 3). *** indicates *p* < 0.001 (2-ways ANOVA). (**E**) Western blots showing TRPM7, p53, CDK2, PCNA, GAPDH expression following treatment with Nutlin-3a. (**F**) Relative protein expressions of TRPM7, p53, CDK2 and PCNA following treatment with Nutlin-3a (*n* = 3–4). * indicates *p* < 0.05 and *** indicates *p* < 0.001 (2-ways ANOVA).

**Figure 4 cells-11-02255-f004:**
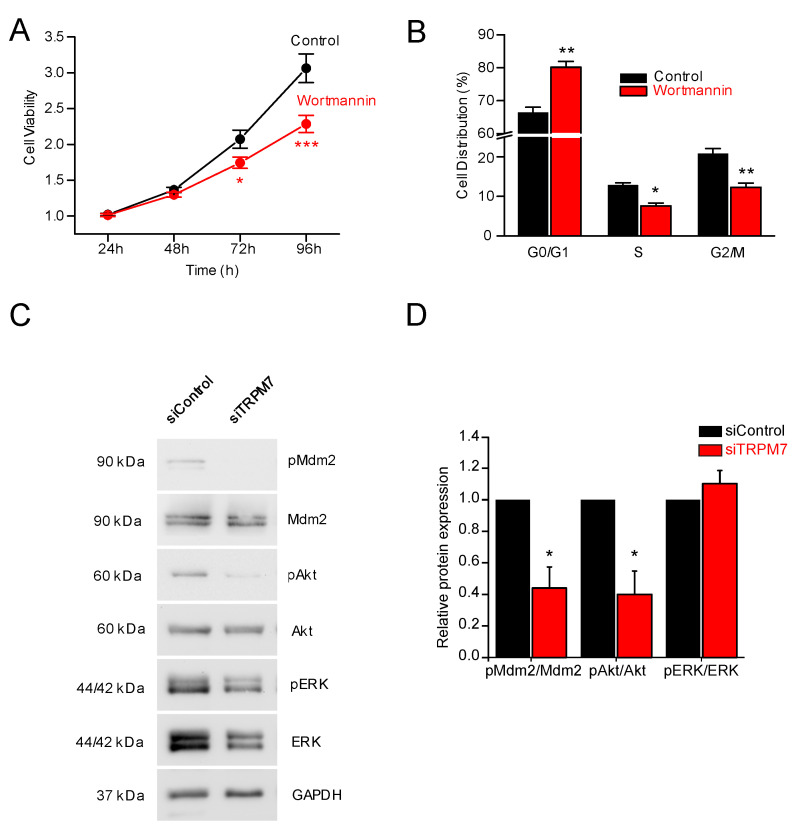
TRPM7 is able to regulate PS-1 proliferation through the PI3K/Akt pathway. (**A**) Cell viability assessed without (Control) or with wortmannin (an inhibitor of the PI3K). * indicates *p* < 0.05 and *** indicates *p* < 0.001 (2-ways ANOVA). (**B**) Following wortmannin treatment, PS-1 cells accumulated in G0/G1 phase, whereas the percentage of cells decreased in S and in G2/M phases (*n* = 3). * indicates *p* < 0.05 and ** indicates *p* < 0.01 (2-ways ANOVA). (**C**) Immunoblotting of PS-1 lysates. Cells were either transfected with a scramble siRNA (siControl) or siTRPM7, and incubated with antibodies against phosphorylated and total forms of MDM2, Akt, and ERK. GAPDH expression is used as normalization control. (**D**) Quantifications of relative protein expressions in PS-1 cells of pMDM2/MDM2, pAkt/Akt, and pERK/ERK ratios (*n* = 4–8). * indicates *p* < 0.05 (Mann–Whitney Rank Sum Test).

**Figure 5 cells-11-02255-f005:**
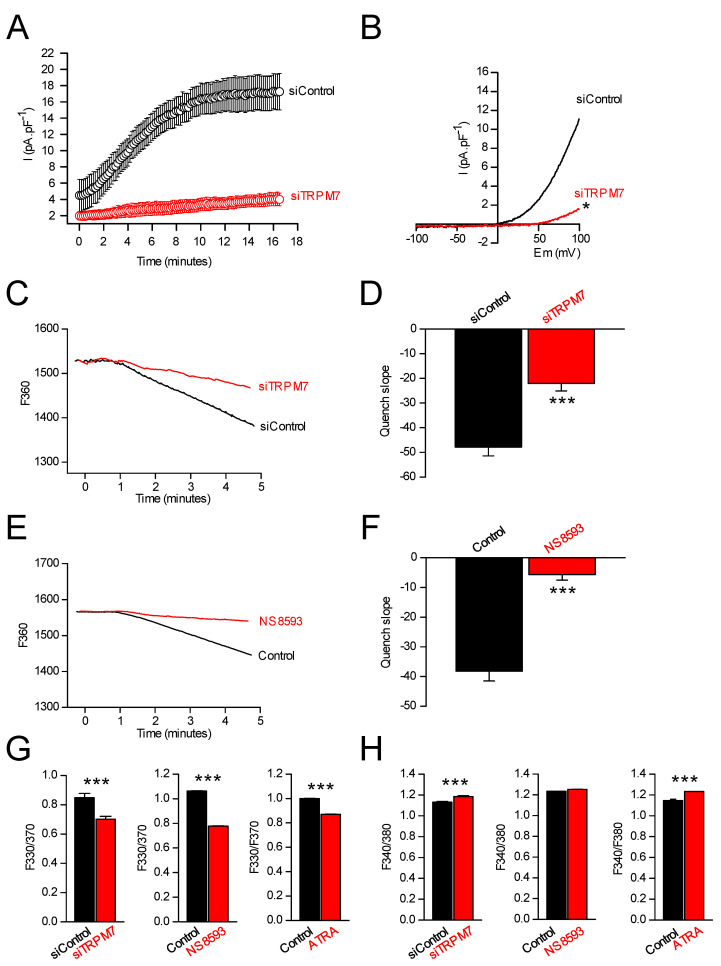
TRPM7 is active and regulates cytosolic Mg^2+^ homeostasis in PSCs. (**A**) Development of an outward Magnesium-Inhibited Cation (MIC) current recorded at +100 mV during the dialysis of intracellular media by a 0-Mg intrapipette solution in PSCs transfected by a control siRNA (*n* = 6) and in PSCs transfected by a siRNA targeting TRPM7 (*n* = 4). (**B**) MIC currents are decreased in PSCs treated by a TRPM7 siRNA (*n* = 4) compared to cells treated by a control siRNA (*n* = 6). * indicates *p* < 0.05 (Mann–Whitney Rank Sum Test for the current-densities recorded at +100 mV). (**C**) Estimation of constitutive divalent cation entry by Mn^2+^-quenching in PSCs treated by a TRPM7 siRNA (*n* = 19) compared to control cells (*n* = 23). (**D**) Quantification of constitutive divalent cation entry after TRPM7 silencing. *** indicates *p* < 0.001 (*t*-test). (**E**) Estimation of constitutive divalent cation entry by Mn^2+^-quenching in PSCs treated by NS8593 (*n* = 34) compared to control cells (*n* = 30). (**F**) Quantification of constitutive divalent cation entry after TRPM7 block. *** indicates *p* < 0.001 (*t*-test). (**G**) Quantification of MagFura-2 fluorescence ratio after treatment with siTRPM7 (*n* = 87 for siTRPM7 and *n* = 92 for siControl), after treatment with NS8593 (*n* = 32 for treated cells and *n* = 39 for control), and after treatment with ATRA (*n* = 40 for treated cells and *n* = 32 for control). *** indicates *p* < 0.001 (*t*-tests). (**H**) Quantification of Fura-2 fluorescence ratio after treatment with siTRPM7 (*n* = 47 for siTRPM7 and *n* = 36 for siControl), after treatment with NS8593 (*n* = 23 for treated cells and *n* = 34 for control), and after treatment with ATRA (*n* = 52 for treated cells and *n* = 27 for control). *** indicates *p* < 0.001 (*t*-tests).

**Figure 6 cells-11-02255-f006:**
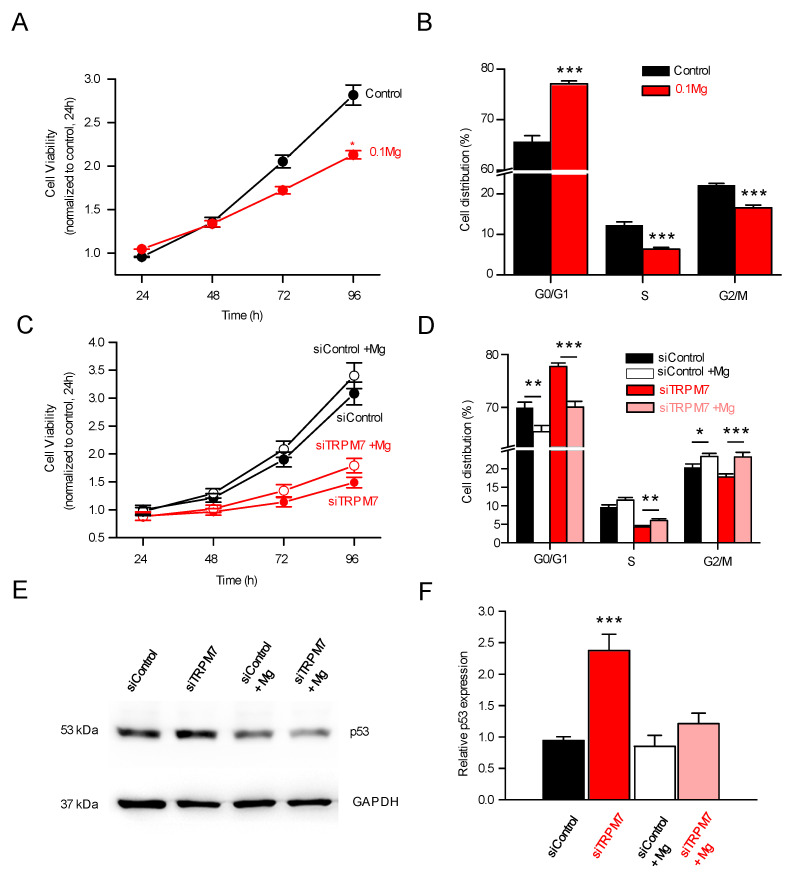
Expression of p53 is magnesium-dependent. (**A**) Cell viability was assessed with usual magnesium concentration (Control) or with reduced magnesium concentration (0.1 mM, *n* = 3). * indicates *p* < 0.05 (2-ways ANOVA). (**B**) When magnesium concentration was reduced (0.1 mM), PS-1 cells accumulated in G0/G1 phase, with a decreased percentage of cells in S and in G2/M phases (*n* = 3). *** indicates *p* < 0.001 (2-ways ANOVA). (**C**) Effect of Mg^2+^ supplementation, with or without TRPM7 silencing, on cell viability (*n* = 3). (**D**) Mg^2+^ supplementation reversed the effect of TRPM7 silencing on cell cycle regulation (*n* = 3). * indicates *p* < 0.05; ** indicates *p* < 0.01; *** indicates *p* < 0.001 (2-ways ANOVA) (**E**) Western blot showing protein expression of p53 in PS-1 cells. Expressions are shown in different conditions: either transfected with siControl or siTRPM7, or in combination with increased magnesium concentrations (72 h post transfection). (**F**) Quantifications of relative p53 expression in PS-1 cells: magnesium supplementation was able to restore the effect of TRPM7 silencing on p53 expression (*n* = 4). *** indicates *p* < 0.001.

**Figure 7 cells-11-02255-f007:**
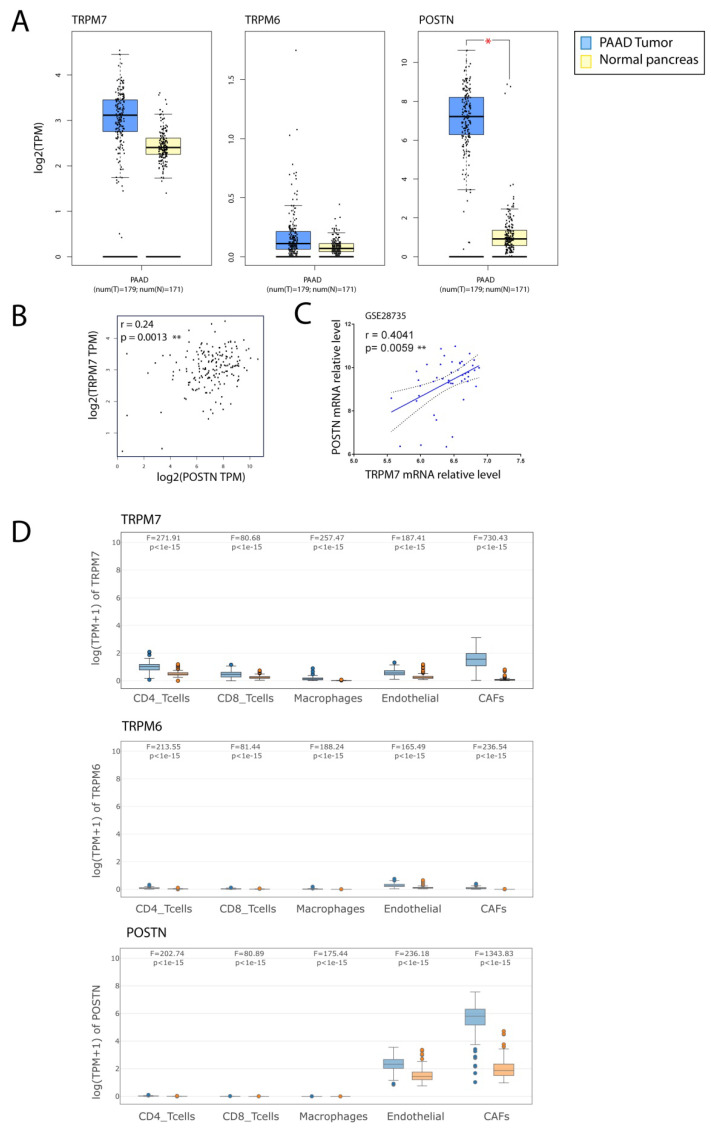
TRPM7 and POSTN expressions are increased in Cancer-Associated Fibroblasts (CAFs) in pancreatic cancer. (**A**) Relative mRNA levels of TRPM7, TRPM6 and POSTN in PDAC and a normal pancreas. Boxplots were generated with GEPIA from TCGA and GTEX datasets. Relative levels are expressed as log2 transcripts per million bases (TPM). * indicates *p* < 0.05 (*t*-test). (**B**) Correlation analysis of TRPM7 and POSTN in PAAD TCGA dataset. ** indicates *p* < 0.01 (*t*-test) (**C**) Correlation analysis of TRPM7 and POSTN in GSE28735 dataset. ** indicates *p* < 0.01 (paired *t*-test). (**D**) Relative mRNA levels of TRPM7, TRPM6 and POSTN in CD4+, CD8+, endothelial cells, macrophages and cancer associated fibroblasts (CAFs) from TCGA tumors and normal pancreas samples. EPIC deconvolution of the dataset was performed using GEPIA2021 tool. Quantitative comparisons of TRPM7, TRPM6 and POSTN expression were analyzed using ANOVA test.

**Figure 8 cells-11-02255-f008:**
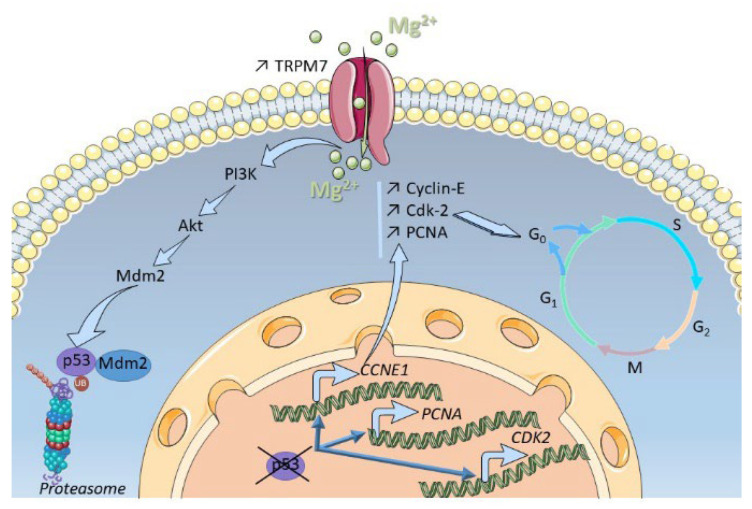
Overview of the pathway activated by TRPM7 and magnesium in PSCs.

**Table 1 cells-11-02255-t001:** Antibodies used in Western blotting.

Primary Antibody	Supplier	Dilution	Species
TRPM7	Abcam (ab109438)	1:1000	Rabbit
Vimentin	ThermoFisher (MA5-16409)	1:1000	Rabbit
pERK1/2	Cell Signaling (9101S)	1:1000	Rabbit
ERK1/2	Cell Signaling (9102S)	1:1000	Rabbit
pAkt	Cell Signaling (9271S)	1:250	Rabbit
Akt	Cell Signaling (9272S)	1:500	Rabbit
pMdm2	Abcam (ab170880)	1:1000	Rabbit
Mdm2	Abcam (ab38618)	1:1000	Rabbit
PCNA	Abcam (ab29)	1:1000	Mouse
Cyclin-E1	Abcam (ab133266)	1:500	Rabbit
CDK2	Cell Signaling (2546S)	1:500	Rabbit
p53	Santa Cruz (sc126)	1:500	Mouse
GAPDH	Abcam (ab8245)	1:4000	Mouse
HRP-linked Anti-rabbit IgG	Cell Signaling (7074S)	NA	Goat
HRP-linked Anti-mouse IgG	Cell Signaling (7076S)	NA	Horse

## Data Availability

TCGA, GTEX and GSE data are available and are based upon public data extracted from TCGA Research Network. Available online: http://cancergenome.nih.gov/ (accessed the 2 March 2022), Genome Tissue Expression (GTEX) project. Available online: http://www.GTEXportal.org/ (accessed the 2 March 2022), and Gene Expression Omnibus (GEO) database. Available online: http://www.ncbi.nml.nih.gov/geo/ (accessed the 2 March 2022).

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
