# Peer review of "TRPM7 Modulates Human Pancreatic Stellate Cell Activation"

_cells, 2022, doi:10.3390/cells11142255_

Round 1

Reviewer 1 Report

This is an interesting but rather scrambled up MS elucidating role of TRPM7 in pancreatic stellate cell physiology and possible involvement in pancreatic pathology.  The MS is rather unfocused and complex. Two cell lines of the pancreatic stellate cell origin were used. The PS-1 was from normal pancreas, TLR-PSC was from pancreas showing chronic pancreatitis. No systematic comparison was made between the two cell lines which is strange. Most works were done in the normal PS-1 cell line. PS-1 showing low TRPM7 expression was knocked down in TRMP7 expression, instead of doing over-expression. TLR-PSC showing high levels of TRMP7 expression was not used for such TRMP7 knock down experiments.

MS must be validated in the following respects. 1/  Please clarify what cell viability means (changed metabolic rate / cell proliferation?) in the MS. 2/ Please clarify clearly the Mg2+ gradient across the plasma membrane. 3/ Please clarify whether you aim to investigate role of pancreatic stellate cells in chronic pancreatitis or in pancreatic cancer. 

Minor points

1/ Introduction:

1.1 Concise summary should be provided of the role of pancreatic stellate cells on pancreatic exocrine cells.

1.2 Clarify that TRPM7 is a two-domain protein, briefly summarize function of each domain.

1.3 Concisely summarize Mg2+ transport mechanism, including concentrations of the Mg2+ gradient across the plasma membrane. 

1.4 Clarify aim of study: chronic pancreatitis or cancer? If both, then MS is rather unfocused?  

2/ Methods

2.1 Are all cell lines listed in Materials and Methods? Please include cancer cell line PACA (PACA-conditioned medium was used).

2.2 "All chemicals purchased from Sigma" is probably not true. Please specify. ALL specific reagents should be listed with City and Country of provider, with Cat. No. 

2.3 The meaning of the so-called "Viability Assay" is NOT clarified. Please specify whether you are measuring cell number / cell proliferation, or cell metabolism. Modify the text description accordingly.

2.4 Please clarify how to distinguish cytosolic Mg2+ from ER Ca2+ signal when using Mag-Fura-2. Or whether such consideration is not needed. 

3/  Results

3.1 Figure 1: Clarify source of cancer cell medium, or please list this cell line in Materials and Methods. Why not use conditioned medium from RLT-PSC, and why not compared TRPM7 expression in PACA-2 cancer cells?

3.2 Figure 2: The rationale of Figure 2 is unclear. Since TRPM7 is expressed at low level in PS-1 cells, why down regulate TRPM7 expression in PS-1 cells but not in RLT-PSC cells. If one needs to elucidate effect of TRPM7, TRPM7 probably needs to be over-expressed in PS-1 cells. Main text Lines 213-215: TRPM7 silencing or inhibition?

3.3 Figure 3: Effects of P53 inhibition should be corroborated with P53 silencing?

3.4 Figure 4D: These three graphs could be combined (on the same x and y axis). 

3.5 Figure 5G, H: Please present typical original tracings or records.

3.6 Figure 6C: Curves should be marked more clearly. 

3.7 Figure 7D: Please give actual P values. 

3.8 Please clarify how to distinguish the different cell types in tumor or normal tissue samples. The cell type-specific data or the origin of such data are unconvincing in the present version of the MS. 

3.9 Others: In text in Results, "cell distribution" > "cell cycle distribution"? Line 366: What is "sub-expression"? 

4/ Discussion   

Line 447: Mg2+ entry via TRMP7 in PS-1 cells: But clearly outward currents were detected in this work?

Figure 8: Please show clearly the direction of Mg2+ movement across the plasma membrane. 

Reviewer 2 Report

The paper by Auwercx et al contains data that add to the picture already provided in the published literature, indicating a role for TRPM7 in pancreatic stellate cell activation.

[1] One weakness of the paper is the rather minimalistic description and documentation relating to the patch clamp data. It would have been comforting to see some real current traces and also have some controls regarding effects or no effects of the procedures used on a channel type other than TRPM7.

[2] Another weakness relates to the data on the apparent intracellular Mg2+ concentrations (Fig. 5G,H). Although it is stated that the differences between the black and red columns are statistically significant, the actual difference in most of these experiments is really VERY small, making it doubtful whether there is any real biological significance here.

[3] The Introduction or Discussion should include some cautionary remarks with regard to the interpretation of the results as they are all based on cell lines. As previously shown (Ferdek et al J Physiol 2016), there are major differences in the behavior of real pancreatic stellate cells and stellate cell lines.  

[4] Finally, it would make the paper more generally interesting if the results were briefly put into the context of real (intact pancreas, in vivo) pancreatic pathology. Recent data on the interplay between stellate cells, immune cells and acinar cells in the development of pancreatitis (see recent major review by Petersen et al, Physiol Rev 2021), which is a major risk factor for – and often precursor of - pancreatic cancer, are highly relevant.

Round 2

Reviewer 1 Report

MS investigated role of TRMP7 in normal human pancreatic stellate cell (line) activation, with implications for chronic pancreatitis and pancreatic cancer. The main narrative of the MS could be more focused and streamlined. Relevance of dry data on cancer-associated fibroblasts is uncertain. MTT assay might be complemented by simple counting of cell numbers. The following points need to be addressed.

1/ In INTRODUCTION, description on pancreatic stellate cell modulation of pancreatic acinar cell function is rather incomplete, and at least unbalanced.

The authors stated that "Thus, PSCs may regulate exocrine function through the cholecystokinin-induced release of acetylcholine [5]".

The quoted work (Proc Natl Acad Sci USA 107: 17397-17402, 2010. doi: 10.1073/pnas.1000359107) suggested that PSC could possibly release some ACh to elicit amylase secretion from isolated pancreatic acini. But the amount of amylase release was rather miniscule and negligible, from 100% to 120% of basal level (normal maximal stimulation could reach 4-6 fold of basal).

On the other hand, compelling evidence indicates that PSC could actually completely block amylase secretion or associated calcium signaling in the pancreatic acinar cells:

Figure 2C, Pancreatology 16: 570-577, 2016. doi: 10.1016/j.pan.2016.03.012; Figure 5, Cells 8: 109, 2019. doi: 10.3390/cells8020109.

2/ In RESULTS, Figure 6C: The lettering is rather crowded and remains unclear and confusing. Please try to spread the letterings to both sides of the curves for clarity, or leave some distance in between the labelings.

Figure 7 is of exceptionally low resolution and poor quality. Figure 7D: TRP7, TRPM6, POSTN: The P value of less than 1e-15 is repeated multiple times. If the P values are different, please place the actual value above each column. Judging from the size of the SEM, surely the P values were not all that small (less than ten to the minus sixteenth power?).

3/ Mg2+ flux

Data indicate that TRPM7 probably mediates Mg2+ efflux instead of influx. Both free and anion-bound Mg2+ in the cytosol are higher or much higher than in extracellular medium, both by consensus and from the authors own previous publications. Mg2+ current may not all be inward, just like chloride current might not all be outward depending on the cytosolic chloride concentration. Please elaborate more on this point in the main text, possibly to attract more audience / readers and hopefully more citations.

4/ Could you discuss the possible contribution by ER calcium (or no contribution) to the fluorescence of Mg2+-Fura-2, even under basal and un-stimulated conditions?

5/ Since the authors put a lot of emphasis on the chronic pancreatitis cell line, conditioned medium from that cell line should be applied to the normal PSC cell line for comparison (data to be added to Figure 1F).

6/ I still believe for cell proliferation assay, MTT is not appropriate; one only needs to count the cell numbers at different points of culture. MTT could mean either dead cells or decreased cell numbers, or both, and is more suitable for cytotoxicity assays.

7/ Line 407: Please explain or revise the phrase “sub-expression”.

Reviewer 2 Report

The revised version looks OK.

Author Response

We thank the reviewer for his comment.